# Rhythm and groove as cognitive mechanisms of dance intervention in Parkinson's disease

**Anna Krotinger**[ID][1,2☯], **Psyche Loui**[ID][1,3,4☯]*

**1** Department of Biology, Wesleyan University, Middletown, Connecticut, United States of America, **2** Center for Bioethics, Harvard Medical School, Boston, Massachusetts, United States of America, **3** Department of Music, Northeastern University, Boston, Massachusetts, United States of America, **4** Department of Psychology and Program in Neuroscience and Behavior, Wesleyan University, Middletown, Connecticut, United States of America

☯ These authors contributed equally to this work.
* p.loui@northeastern.edu

**Data Availability Statement:** The data underlying the results presented in the study are available from figshare: 10.6084/m9.figshare.13034165.

**Funding:** This project is supported by Grammy Foundation, Kim & Glen Campbell Foundation, and

## Abstract

Parkinson's disease (PD) is associated with a loss of internal cueing systems, affecting rhythmic motor tasks such as walking and speech production. Music and dance encourage spontaneous rhythmic coupling between sensory and motor systems; this has inspired the development of dance programs for PD. Here we assessed the therapeutic outcome and some underlying cognitive mechanisms of dance classes for PD, as measured by neuropsychological assessments of disease severity as well as quantitative assessments of rhythmic ability and sensorimotor experience. We assessed prior music and dance experience, beat perception (Beat Alignment Test), sensorimotor coupling (tapping to high- and low-groove songs), and disease severity (Unified Parkinson's Disease Rating Scale in PD individuals) before and after four months of weekly dance classes. PD individuals performed better on UPDRS after four months of weekly dance classes, suggesting efficacy of dance intervention. Greater post-intervention improvements in UPDRS were associated with the presence of prior dance experience and with more accurate sensorimotor coupling. Prior dance experience was additionally associated with enhanced sensorimotor coupling during tapping to both high-groove and low-groove songs. These results show that dance classes for PD improve both qualitative and quantitative assessments of disease symptoms. The association between these improvements and dance experience suggests that rhythmic motor training, a mechanism underlying dance training, impacts improvements in parkinsonian symptoms following a dance intervention.

## 1. Introduction

Parkinson's disease (PD) is a neurodegenerative disorder characterized by motor symptoms including tremor, rigidity, and akinesia, which affect daily activities such as walking and speaking [1]. The cognitive mechanisms underlying the motor symptoms likely involve a loss of internal cueing systems, which are evident from motor tasks such as walking, finger tapping, and musical rhythm perception [2–5]. These tasks all depend on rhythmic timing, either

National Science Foundation NSF-CAREER 1945436 to PL. The funders had no role in study design, data collection and analysis, decision to publish, or preparation of the manuscript.

**Competing interests:** The authors have declared that no competing interests exist.

emergent or event-based, through rhythmic movement or entrainment to an external cue [6]. The perception of rhythmic timing and subsequent production of rhythmic movement are dependent on dopaminergic activity in the corticostriatal circuits [7–9], including dopaminergic cell loss in the substantia nigra pars compacta, resulting in diminished levels of available dopamine [10]. As a result, PD patients show decreased activation of the basal ganglia when listening to music, as compared with healthy adults [11], in addition to deficits in both rhythmic perception and production [6,11–14].

Music- and dance-based interventions, which provide external auditory and visual cues, have been explored as a means of supplementing the timing and cueing deficits resulting from basal ganglia impairment in PD [15–19]. One specific type of intervention is Rhythmic Auditory Stimulation (RAS), in which participants walk to an external auditory stimulus such as a metronome or a song to supplement rhythmic cueing [20,21]. PD patients after training with RAS showed improvements in gait and stride length [20,21], which in turn decrease rates of falling as a result of providing external, rhythmic support [22].

Dance is an activity that relies heavily on music, specifically on entrainment to musical rhythm, and combines rhythmic auditory and visual cues to coordinate movement. Several case studies, quasi-randomized pilot trials, and meta-analytic evidence have shown beneficial effects of dance on both motor and non-motor PD symptoms including balance, gait, quality of life, mood, attention, and memory [19,23–27]. These findings have inspired the development of dance programs for PD [28]. Dance interventions, such as classes in Argentine Tango, jazz, or classical ballet, have specifically been shown to improve both gait and balance in individuals with PD [15,29–32]. Notably, while patients with PD exhibited improvements in balance following a dance intervention, these improvements were not observed in PD patients who had completed a rote exercise program [24], suggesting a unique role for dance in assuaging parkinsonian symptoms. These findings are clinically important as problems with gait and balance are both very common in PD, and are also associated with increased rates of falls and other adverse events that decrease quality of life in older adults more generally [33]. Because of their effectiveness, dance intervention has been posited to benefit even other older adults beyond those with PD [34].

## 1.1 Individual differences in neural structure and function impact rhythm perception and production

The neural mechanisms supporting these benefits are likely related to the effects of music and dance on brain structure and function, and while the evidence supporting the beneficial impact of music- and dance-based interventions on PD is strong, responses to these interventions may vary based on individual differences in rhythm perception and production. Dalla Bella, et al.'s report [21] that RAS improved gait in PD patients also found that the degree of improvement was dependent on patients' performance during a simple tapping task. Thus, the ability to entrain or couple movements to music—sensorimotor coupling ability—was a predictor of therapeutic benefit from RAS. Furthermore, this finding suggests that objective rhythm tests, such as a simple tapping test to assess sensorimotor coupling ability, may be associated with improvements in PD symptoms.

Sensorimotor experience and ability in healthy adults also affect individual differences in neural structure and function. Having received formal training in music, for example, has been found to impact grey matter structure in the premotor and supplementary motor areas [35–39]. People with musical training also show superior beat perception compared with those without musical training, with concomitant underlying differences in functional connectivity between premotor and striatal areas [39].

Response to music is also impacted by subjective experience: while the tendency to move one's body when music is playing is often unconscious, the degree of spontaneous movement to a certain piece of music may depend on a combination of musical or acoustic factors (syncopation, tempo) and subjective measures (enjoyment, familiarity). Distinct responses to different aspects of rhythm contribute to the perception of temporal structure in music. As these patterns typically obey a regular structure, the progression of a given rhythm is predictable [40]. Interestingly, there is evidence to support the notion that the ability to predict a sequence of beats or the rhythmic progression of a melody is related to the extent to which a listener enjoys a certain piece of music [37]. In this sense, the perception of rhythm, meter, and beat are important not only for music processing, but also for emotional and psychological responses to music, which is shown to drive sensorimotor coupling [41,42]. Evidence for the role of groove in sensorimotor coupling comes from results involving transcranial magnetic stimulation (TMS) of the motor cortex: when asked to tap along to different songs, healthy participants show the most spontaneous motor excitability, as indicated by motor evoked potentials in response to TMS, in response to high-groove songs, as compared with songs categorized as low-groove [43]. This suggests that groove mediates the way auditory rhythms excite the motor system; in other words, groove drives sensorimotor coupling. Applied to PD, this suggests that groove may be a factor that can influence responsiveness to dance interventions, due to its effect on spontaneous motor excitability.

Dance experience has also been related to differences in patterns of neural activation, and is found to engage brain areas implicated in movement, especially rhythmic movement. When dancers observe others dancing, they show greater activity in the frontoparietal action observation network (AON), a circuit involved in the observation and production of movement [44]. While the AON is consistently activated when watching others demonstrate movements [19], the degree of AON activation is related to the familiarity of the task being observed, suggesting experience-dependent specificity [41]. Dancers have also shown increased functional connectivity in cortico-striatal pathways that are implicated in posture, movement, and action selection [45], enhanced white matter diffusivity [46], and enhanced neuroplasticity in motor regions [47]. These findings suggest that dance experience impacts the structure and function of brain regions involved in both the observation and production of movement.

Training in both music and dance, then, have measurable effects on the structure and function of brain regions involved in rhythm perception and production. However, it is still unclear how these experiences, in addition to variations in musical groove, might impact the effect of dance-based interventions on PD patients.

## 1.2 Current study and hypothesis

The current study aims to assess and predict the effects of four months of dance classes on parkinsonian symptoms. Specifically, we examine factors that could influence individual differences in responsiveness to dance classes. While studies have demonstrated that dance classes for PD benefit some disease symptoms, they have not assessed improvement in PD symptoms in conjunction with objective tests of rhythmic behavior. Rhythmic tapping behavior was previously found to be associated with the therapeutic benefit of auditory cueing in a PD population [21]. Here, we ask whether these associations are preserved in the setting of dance classes. We investigated the effects of four months of weekly dance classes for PD on disease severity using the Unified Parkinson's Disease Rating Scale (UPDRS). We further assessed sensorimotor experience, musical groove, and prior experience with music and dance in a group of PD participants and control participants with no PD, in order to determine whether these factors influence the therapeutic outcome of dance classes. In order to assess rhythm capabilities at

different levels of sensorimotor coupling in PD participants, we expanded the tapping task to include varying levels of groove, and added the Beat Alignment Test (BAT) to test beat perception and identification [48]. To confirm that these assessments were valid measures of rhythm capabilities and sensorimotor coupling, we also compared the BAT task and tapping data to an age-matched control group without PD. To our knowledge, there are no normed data from groove-dependent finger-tapping assessments within the age range of those in our PD sample. With this study design, we tested two hypotheses. First, we hypothesize that dance classes for PD improve disease symptoms. Thus, we expect that the symptoms of PD participants would reduce as assessed by the UPDRS after dance classes. Second, we hypothesize that individual differences in dance experience, rhythmic ability, and sensorimotor coupling would all affect the therapeutic outcome of these classes. To test this hypothesis we assessed dance experience, rhythmic ability using the BAT, and sensorimotor coupling using tapping to high-groove and low-groove songs, in both non-PD controls as well as in PD participants before and after intervention. Fig 1 provides a conceptual overview of the relationships tested in the present study.

## 2. Materials and methods

### 2.1 Participants

**2.1.1 Recruitment.** All participants provided written informed consent as approved by the Institutional Review Board of Wesleyan University, which approved this study (protocol number 20180420-akrotinger-pd). Participants, ranging in age from 59–84 (n = 30), were recruited from PD dance classes in New York City, NY (n = 18), San Rafael, CA (n = 7), and

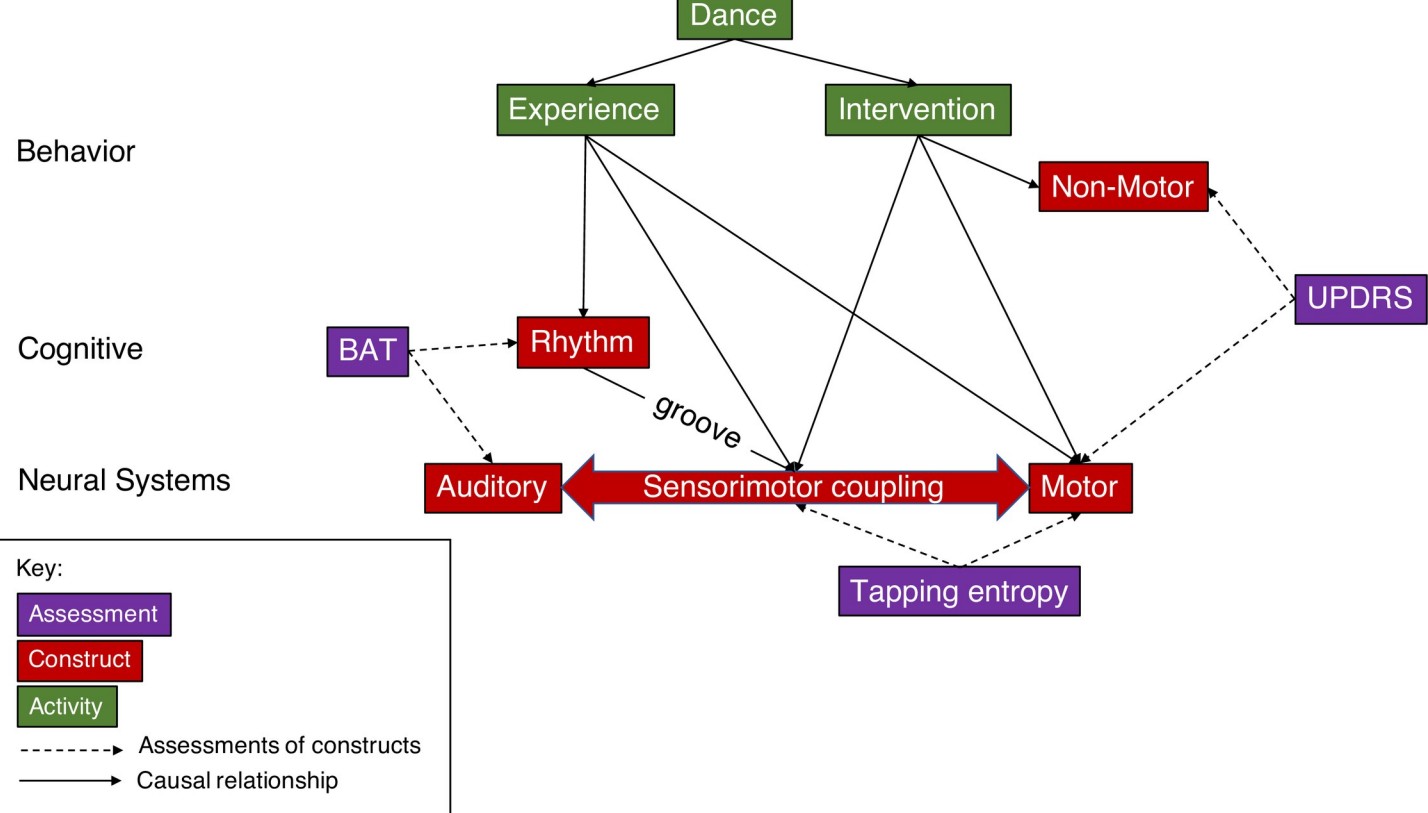

**Fig 1. Conceptual overview of relationships between relevant cognitive and behavioral assessments, constructs, and activities in the context of Parkinson's disease.**

Santa Rosa, CA (n = 5), and included 7 men and 23 women. Members of the control group were recruited from acquaintances and spouses of PD participants and included 8 men and 11 women. Control participants (n = 19) were matched for age, handedness, music experience (ME), and dance experience (DE). Participants were considered to have ME or DE if they had over one year of formal training in some form of music or dance, respectively.

**2.1.2 Intervention.** All dance classes were taught by instructors with teaching certification from Dance for PD®. While the specific content of each class changed weekly and teachers were free to draw on different dance styles (e.g. African or Latin) while planning choreography, the structure was consistent and the primary form of movement was in a contemporary/modern style. Each class lasted 75 minutes and was set to live music. About 40 minutes were spent seated. While seated, instructors led exercises emphasizing muscle and limb extension and rhythmic movement. These exercises were typically broken into combinations focused on coordination (dancers might be instructed to clap on certain counts, or to alternate clapping and stomping), footwork (involving leg extension, flexing and pointing the feet, and raising heels to a forced arch position, for example), and torso (focusing on leaning forward and side to side, stretching the sides of the body, and coordinating arm movements with other motions), and explore a variety of musical tempos and rhythms.

For the remainder of the class, instructors invited participants to stand if they were able. For about twenty minutes, participants performed exercises standing at the barre or holding on to the back of their chairs. These combinations emphasized plié, or bending and straightening the knees; lifting the heels off the ground to balance; and extending the legs and feet in different directions. Finally, participants moved across the floor for the last fifteen minutes of class, performing exercises involving weight shifting, marching, partnering (participants mirror others' movement), and improvisation. In preparation for each combination, the instructor demonstrated a sequence of movements and then provided spoken cues while all participants performed the combination to music. Each class had at least one volunteer who provided dancers with support and assistance.

**2.1.3 General design (Pre- & post-intervention sessions).** All 30 PD participants and 19 control participants completed the pre-intervention interview. This interview included four tasks: a questionnaire, the UPDRS (PD participants only), the BAT, and a tapping task. 20 PD participants completed the post-intervention interview, which included the UPDRS, the BAT, and the tapping task. Post-intervention data could not be collected in three participants because of death (n = 2) and a broken hip (n = 1). Five participants were unable to be contacted post-intervention, resulting in further loss to follow-up. Finally, two participants failed exclusion criteria for the post-intervention interview, so their data was not included in post-intervention analyses. There was no significant difference between baseline UPDRS scores for PD participants who did and did not complete the post-intervention interview (t(11.5) = 1.30, p = 0.218), indicating that there was no self-selection for post-intervention testing.

**2.1.4 Inclusion/Exclusion criteria.** Inclusion criteria for members of the experimental group included a Parkinson's diagnosis, consistent weekly attendance of PD dance classes, and the ability to perform the tasks included in the study. Participants either had to be free of hearing impairments or, if they did have impairment, use a hearing aid during the study. Participants also had to be able to tap with one finger to complete the tapping task. Exclusion criteria for the post-intervention interview included the initiation of a new pharmaceutical or surgical intervention during the four-month course of the study, and the discontinuation of dance classes between the pre- and post-intervention interviews. If by the post-intervention meeting participants had failed to attend at least one class per week, their data were excluded from follow-up analysis. The minimum requirement for dance class attendance, then, was one class per week, or 16 total classes. We did not control for additional class attendance.

One participant failed the first exclusion criterion as a result of implementing new deep brain stimulation (DBS) settings and experiencing improved motor ability as a result. One participant failed the second criterion, having stopped taking dance classes following the baseline interview.

## 2.2 Procedure

**2.2.1 Neuropsychological assessment battery.** All participants completed a questionnaire on age, location, gender, handedness, hearing impairment, length of time on carbidopa/levodopa, other medications taken to treat PD, most recent dose of carbidopa/levodopa, music experience, and dance experience.

**2.2.2 UPDRS.** The UPDRS was completed by all PD participants and measured disease severity through assessments of both motor and non-motor symptoms. Total scores for the UPDRS range from 0–199, with a score of 0 indicating no disability and a score of 199 indicating the most severe disability. Each question is scored from 0–4. The UPDRS is divided into four sections: Section I measures Mentation, Behavior, and Mood and is scored out of a possible 16 points; Section II measures Activities of Daily Life and is scored out of 52 points; Section III, the Motor Examination, is scored out of 108 points; and Section IV measures Complications of Therapy and is scored out of 23 points.

PD participants completed the UPDRS during both the pre- and post-intervention interviews. For participants on carbidopa/levodopa who experienced distinct "on" (more active) and "off" (less active) phases due to medication cycles, they were always in the "on" phases during the assessments. No video recordings of the UPDRS were taken because participants did not consent to being videotaped.

**2.2.3 BAT.** The BAT tests beat perception, capturing individual differences in beat processing. For the current study, twelve music samples were selected from the Iversen and Patel paper [45] and played for all participants. A beep track was overlaid on the music in each track after five seconds. The onset of the beep track was timed so that it either directly coincided with the underlying beat (same phase and same frequency), or was 10% faster or slower than the rhythm of the music. Of the twelve music samples, five were "on the beat" and seven were "off the beat" (S1 Table). Participants were asked to identify whether the beep track was on or off the beat and to rate their confidence in each answer.

**2.2.4 Tapping task.** Participants completed a simple tapping task to assess sensorimotor coupling ability. Eight songs were selected from Janata, Tomic, and Haberman's groove index [41], which quantified musical groove on a scale of 0–127: "Cheek to Cheek", "In the Mood", "Sing Sing Sing", "Superstition", "Carolina in my Mind", "Comfortably Numb", "'Til There was You", and "What a Wonderful World." The first four songs listed were designated as "high-groove" (Mean groove index = 97.4, SD = 9.4) and the last four as "low-groove" (Mean groove index = 52.0, SD = 10.2). All "high-groove" songs were similarly designated as high-groove per Janata et al., and all the "low-groove" songs were originally designated as either mid or low-groove (S2 Table). Each song was imported into GarageBand version 10.3.2 on a MacBook Pro and edited to a 30 second excerpt.

A KORG nanoPAD2 used to record participants' tapping. Participants were instructed to use one finger on their dominant hand to tap along to the beat of the music sample. They were asked to begin tapping as soon as they identified a beat of their choosing, and to continue tapping until the music stopped. The tapping track was then isolated and converted to a Wav file for analysis. After tapping to each excerpt, participants were asked to rate their enjoyment of and familiarity with the song on a scale of 1 (least enjoyable, least familiar) to 3 (most enjoyable, most familiar).

## 2.3 Data analysis

**2.3.1 UPDRS.** Data were analyzed in RStudio (version 1.1.456) and MATLAB (The Math-Works, Inc., Natick, MA, version R2018a). Post-intervention UPDRS scores (n = 20) were subtracted from pre-intervention UPDRS scores to determine any changes in disease severity. Z-scores were calculated for each change in UPDRS score to normalize skewed distributions and analyzed using one-way ANOVA and Spearman's correlation. Missing data due to loss to follow-up (n = 10) resulted in fewer degrees of freedom for statistical analyses assessing changes in UPDRS scores than in analyses relying on data from pre-intervention interviews.

**2.3.2. BAT.** False alarm rates (FAR) and hit rates (HR) were calculated from participants' categorizations of musical samples as "on" or "off" the beat. HR is the proportion of correctly detected differences in a set of musical samples given the overall number of differences, whereas FAR was the proportion of trials where the response was "different" given that the correct response was "same." Average FAR (PD = 24.3%, control = 15.0%) and HR (PD = 77.3%, control = 84.3%) were calculated for both experimental and control groups (Results Table 1). Z-scores for HR and FAR were then calculated and participants' FAR z-scores were subtracted from their HR z-scores to get d', a measure of sensitivity. D' was used as a measure of BAT performance, as higher d' values indicated greater sensitivity to changes in stimuli.

**2.3.3 Tapping task.** Audio files of recorded tapping were loaded into Matlab, where the MIDI Toolbox [49] was used to produce a waveform representation of each tapping track. Every waveform was then analyzed to determine the onset of tapping events. Inter-onset intervals (IOI) were computed from tapping events. The first three IOI values for each tapping recording were omitted from analysis to allow a period of time during which participants were becoming accustomed to the task. IOI values > 3 seconds were also omitted from analysis (affecting 59 out of 552 total trials), as values this high were reflective of either a misunderstanding of the task or a failure of the KORG nanoPAD2 to register a tapping event (this affected 12.1% of pre-intervention PD trials, 12.5% of post-intervention PD trials, and 6.6% of control trials). Linear mixed-effects models (LMEs) were constructed using the R function LmerTest::lme4 to investigate whether Groove (High vs. Low), Group (PD vs. Control), and their interaction affected IOI values. First, a base model was constructed using IOI as the dependent variable and participant ID as a random effect. Then, Groove and Group were added as fixed effects. Maximum likelihood (ML) was used for model fitting. Models were compared via likelihood ratio tests using the anova function. All code for analyses and figures are freely available on https://doi.org/10.6084/m9.figshare.13034165.v1.

Shannon entropy, a measure of the randomness of events, was calculated for each trial of the tapping task. This was computed by first extracting a polar histogram, set to 400 bins, from tapping IOIs for each trial to identify the frequency of events at each distinct phase angle. Then, Shannon entropy [50] was computed for the histograms using the formula

$$H(x) = -\Sigma^n_{i=1} \left[ P(x_i) * \log P(x_i) \right] = \Sigma^n_{i=1} \left[ P(x_i) * \log\left(1/P(x_i)\right) \right]$$

**Table 1. Subject characteristics for PD and control groups.**

| Group | | PD | Control |
|---|---|---|---|
| N | | 30 | 19 |
| N with Dance Experience (DE) | | 20 (66.7%) | 12 (63.1%) |
| | Average ± SD length of DE in years | 7.4 ± 7.60 | 16.0 ± 15.1 |
| N with Musical Experience (ME) | | 19 (63.3%) | 11 (57.9%) |
| | Average ± SD length of ME in years | 23.4 ± 22.9 | 20 ± 22.2 |

where $x_i$ represents a single bin (or phase angle) in the polar histogram, and $P(x_i)$ represents the probability of that bin. Shannon entropy reaches 0 if the probability of a certain phase angle is 1. Lower entropy values, then, represent decreased randomness. For the purposes of our study, the highest possible entropy value, indicating total randomness, is equal to $-(400*1/400*\log(1/400)) = 2.602$. The lowest possible entropy, indicating robotic tapping, is 0. Entropy was then analyzed using linear mixed-effects models (LMEs). A base LME model was constructed with log(Entropy) as the dependent variable and participant ID as a random effect. Then, Time-Point (pre- vs. post-intervention), Groove (high vs. low), and Years of Dance Experience (DE) (to assess the impact of discrepancies between average duration of DE in control and PD participants) were added as fixed effects for an initial analysis of differences between pre- and post-intervention PD entropy. To evaluate the potential differences between PD and control entropy separately for each time-point, we constructed models using 1) pre-intervention PD and control entropy values, and 2) post-intervention PD and control entropy values, each of which included Group (PD vs. control) as a fixed effect in addition to Groove and Years of DE. Finally, Presence of DE and Presence of Music Experience (ME) were added as fixed effects to LMEs for a further analysis of pre- and post-intervention PD entropy.

Significance levels for all analyses were denoted as following in the following text and in all figures: * = p < 0.05, ** = p < 0.01, *** = p < 0.001.

## 3. Results

### 3.1 Demographics and neuropsychological assessments

Participants in the PD and control groups were matched for age, handedness, music experience, and dance experience (Table 1). The majority of participants had either previous music or dance experience (PD: n = 25; Control: n = 15), while a minority had both (PD: n = 14; Control: n = 8).

Pre-intervention UPDRS scores ranged from 6 to 77 (Mean = 29.9, SD = 15.4), representing the variability of disease severity and symptoms in our sample (Table 1). Based on these results, participants generally struggled most with activities of daily life, which include speech, salivation, swallowing, writing, cutting food, dressing, hygiene, and walking. To assess how UPDRS scores changed over time, we calculated the difference between the pre- and post-intervention UPDRS scores of the 20 participants who had completed assessments at both time-points. All but two of these participants exhibited either no change in score post-intervention, resulting in a score difference of zero, or an improvement (decrease) in score, resulting in a negative difference (Table 2). A paired t-test comparing pre- and post-intervention UPDRS scores revealed significant differences between time-points (t(20) = 4.81, p = 0.000106***, d = 0.489) (Fig 2). While the greatest number of participants (n = 17) saw an improvement in scores for UPDRS Section III, the Motor Assessment, there was general improvement across all sections of the UPDRS, with no significant differences between improvements across different sections of the UPDRS (S1 Fig: one-way ANOVA and Tukey's HSD post-hoc testing revealed adjusted p values all > 0.05).

### 3.2 Prior music experience benefits beat perception in PD

Results from the BAT revealed that control participants had higher average d' than PD participants (Table 2; Control: 2.29; PD: 1.70, t(43.6) = -2.34, p = 0.0239*, d = 0.657), demonstrating increased sensitivity to changes in beat and supporting previous work suggesting that PD impairs beat perception [3].

D' measures of sensitivity were compared for PD and control groups by ME and DE to assess the effect of these experiences on beat perception (Fig 3). PD participants with ME

**Table 2. Summary of key results from the BAT, tapping task (pre and post intervention), and UPDRS (pre and post intervention).**

| Assessment | Test | PD Participants | | Control |
|---|---|---|---|---|
| **BAT** (M ± SD) | Hit Rate | 77.3% ± 21.6% | | 84.3% ± 14.7% |
| | FA Rate | 24.3% ± 15.6% | | 15.0% ± 11.1% |
| | D' | 1.70 ± 0.962 | | 2.29 ± 0.795 |
| | | **PD Initial** | **PD Follow-up** | **Control** |
| **Tapping Task** Entropy (M ± SD) | Overall | 0.302 ± 0.120 | 0.275 ± 0.117 | 0.241 ± 0.096 |
| | High Groove | 0.294 ± 0.127 | 0.262 ± 0.117 | 0.234 ± 0.111 |
| | Low Groove | 0.311 ± 0.112 | 0.289 ± 0.115 | 0.249 ± 0.079 |
| **UPDRS** | Section I: Mentation, Behavior and Mood | 2.80 ± 1.86 | 2.00 ± 1.18 | N/A |
| | Section II: Activities of Daily Life | 12.2 ± 7.31 | 9.81 ± 5.90 | N/A |
| | Section III: Motor Examination | 11.1 ± 6.58 | 6.38 ± 4.13 | N/A |
| | Section IV: Complications of Therapy | 3.80 ± 3.84 | 3.05 ± 3.09 | N/A |
| | Total | 29.9 ± 15.4 | 21.2 ± 11.8 | N/A |

performed significantly better on the BAT than PD participants without ME [$F_{(1,28)}$ = 5.37, p = 0.0281*, d = 0.878, Fig 3A], while the presence of DE did not significantly affect BAT performance in PD participants [$F_{(1,28)}$ = 0.647, p = 0.428, d = 0.312, Fig 3B].

No significant differences were found between control participants with and without ME [One-way ANOVA: $F_{(1,17)}$ = 0.009, p = 0.962, d = 0.0438, Fig 3C] or between those with and without DE [$F_{(1,17)}$ = 0.080, p = 0.781, d = 0.134, Fig 3D].

### 3.3 Effects of PD, groove, time-point, ME, and DE on tapping

Fig 4 shows the distribution of intervals between tapping onsets, or inter-onset intervals (IOI), for the tapping task. LME models including Groove and Group as fixed effects revealed that including Groove significantly improved model fit, $\chi^2(1)$ = 5150, p < 2.2e-16***, while including Group did not, $\chi^2(1)$ = 1.18, p = 0.279. F-tests using Satterthwaite's method confirmed the significant main effect of Groove [$F_{(1,16640)}$ = 6037, p < 2.2e-16***], indicating higher IOIs for low-groove songs. The main effect of Group was not significant [$F_{(1,46.5)}$ = 1.14, p = 0.291], but there was a significant interaction between Group and Groove [$F_{(1,16640)}$ =

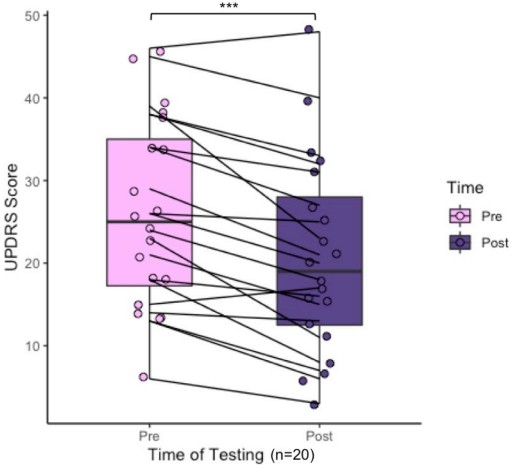

**Fig 2. UPDRS scores pre- and post-intervention for the 20 PD participants who completed both interviews.**

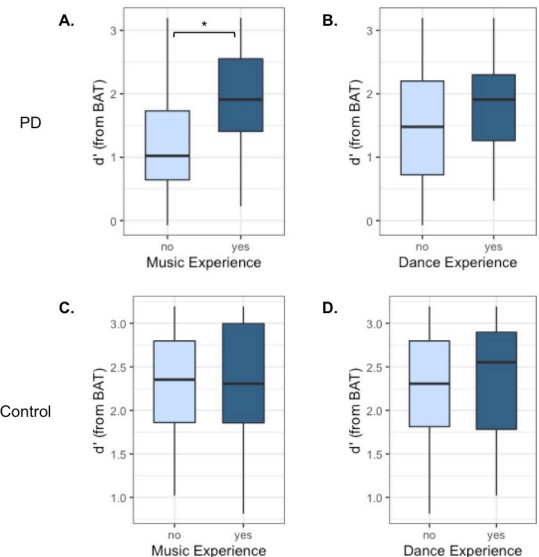

**Fig 3.** BAT results for PD participants with and without ME (A) and DE (B), and for the control group, again with and without ME (C) and DE (D).

20.2, p = 7.07e-06***], indicating that IOI values were higher in the low-groove condition for pre-intervention PD participants only.

We next compared log(Entropy) values of PD participants' tapping pre- and post-intervention (Fig 5A). Greater entropy values reflected increased randomness in tapping events, which we interpreted as a decrease in beat tracking accuracy and sensorimotor coupling ability. Likelihood ratio tests comparing the base model with models including Time-Point, Groove, and Years of DE as fixed effects revealed that including both Time-Point, $\chi^2(1) = 9.75$, p = 0.00179**, and Groove, $\chi^2(1) = 10.6$, p = 0.00113**, independently increased model fit, while including Years of DE, $\chi^2(7) = 12.4$, p = 0.0879, did not. F-tests using Satterthwaite's method confirmed the main effects of Time-Point [F(1,392) = 9.85, p = 0.00183**] and Groove [F(1,378) = 10.7, p = 0.00116**] indicating that post-intervention PD log(Entropy) values were lower than pre-intervention values, and that high-groove entropy values were lower than low-groove entropy (Fig 5B). F-tests revealed no significant interactions between Time-Point and Groove [F(1,374) = 0.0472, p = 0.828].

We then evaluated potential differences between PD and controls in entropy separately for each time-point. For LMEs comparing pre-intervention PD and control log(Entropy) values, likelihood ratio tests revealed that adding Group, $\chi^2(1) = 7.51$, p = 0.00612** and Groove, $\chi^2(1) = 9.48$, p = 0.00208** significantly and independently improved model fit. Adding Years of DE, $\chi^2(11) = 17.1$, p = 0.104, did not improve model fit. F-tests confirmed the main effects of Group [F(1,47) = 7.79, p = 0.00757**] and Groove [F(1,342) = 9.58, p = 0.00213**], indicating that pre-intervention PD log(Entropy) values were significantly higher than control values, and again that high-groove entropy values were lower than low-groove values. All interaction terms were non-significant (all p values > 0.2).

LMEs comparing post-intervention PD and control log(CV) values revealed that including Groove, $\chi^2(1) = 12.6$, p = 0.000393***, but not Group, $\chi^2(1) = 1.88$, p = 0.17, significantly improved model fit. F-tests revealed a significant main effect of Groove [F(1,279) = 12.8, p = 0.000408***], indicating again that high-groove entropy values were lower than low groove values. While including Years of DE also improved model fit, $\chi^2(11) = 22.2$, p = 0.0228*, an F-

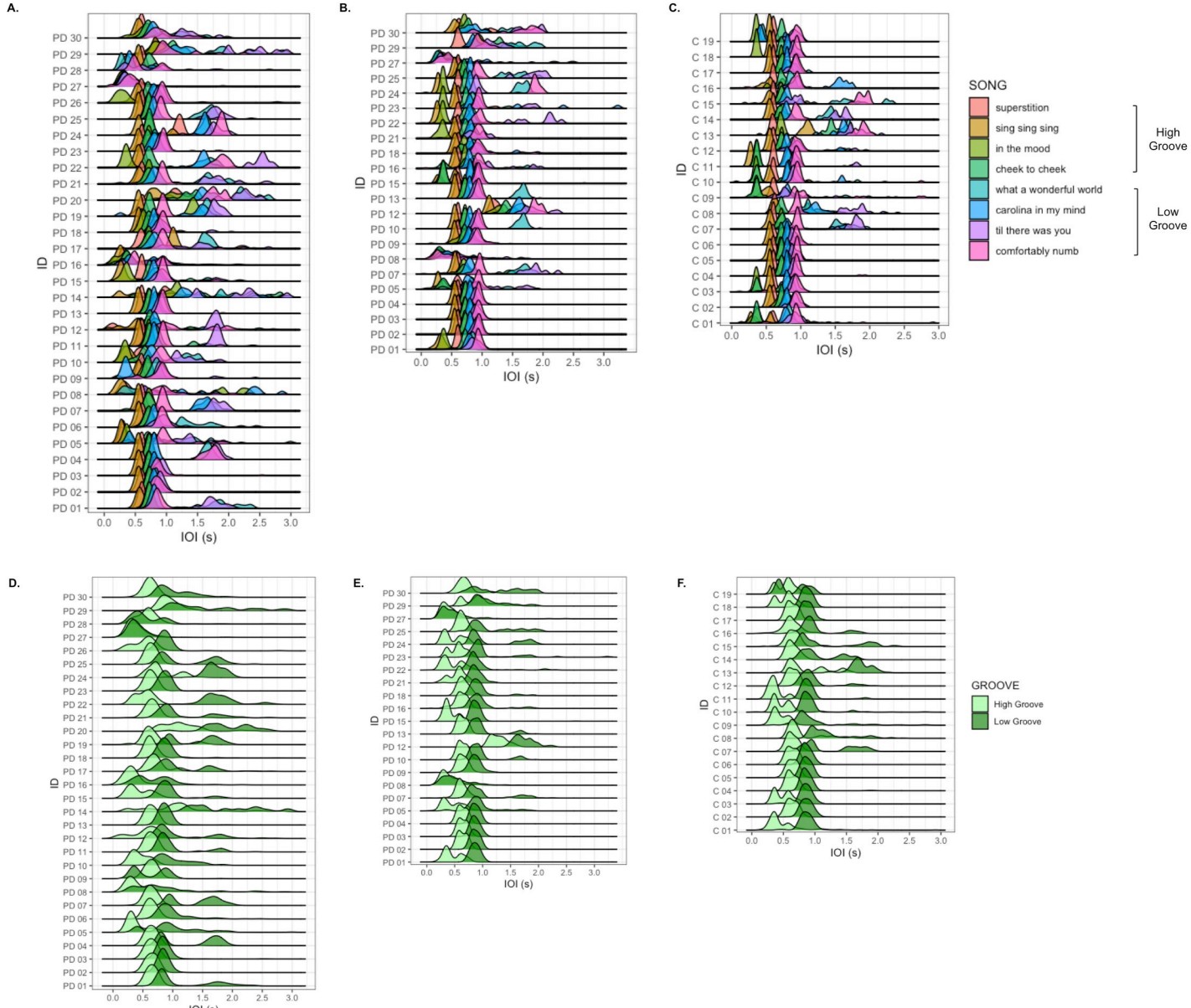

**Fig 4. A.** Distribution of pre-intervention PD inter-onset intervals (IOI) for tapping to each song. **B**. Distribution of post-intervention PD IOI values. **C.** Distribution of control IOI values. **D.** Pre-intervention PD IOI distributions by groove. **E.** Post-intervention PD IOI distributions by groove. **F.** Control IOI distributions by groove.

test did not confirm its independent effect on log(Entropy) [F(1,11) = 1.89, p = 0.0853]. All interaction terms were non-significant (all p values > 0.2). Together, these results demonstrate that PD participants reduced their tapping variability after the intervention, with tapping entropy more similar to that of controls. Furthermore, while tapping entropy varied across songs, high-groove entropy was consistently lower than low-groove entropy.

We then analyzed log(Entropy) values for PD participants by adding Groove, Time-Point, Presence of DE, and Presence of ME, as fixed effects to the base LME model (Fig 5C and 5D). Adding Time-Point, $\chi^2(1) = 9.75$, p = 0.00179**, Groove, $\chi^2(1) = 10.6$, p = 0.00113**, and DE, $\chi^2(1) = 9.02$, p = 0.00267**, to the base model improved model fit; adding ME, $\chi^2(1) = 0.339$,

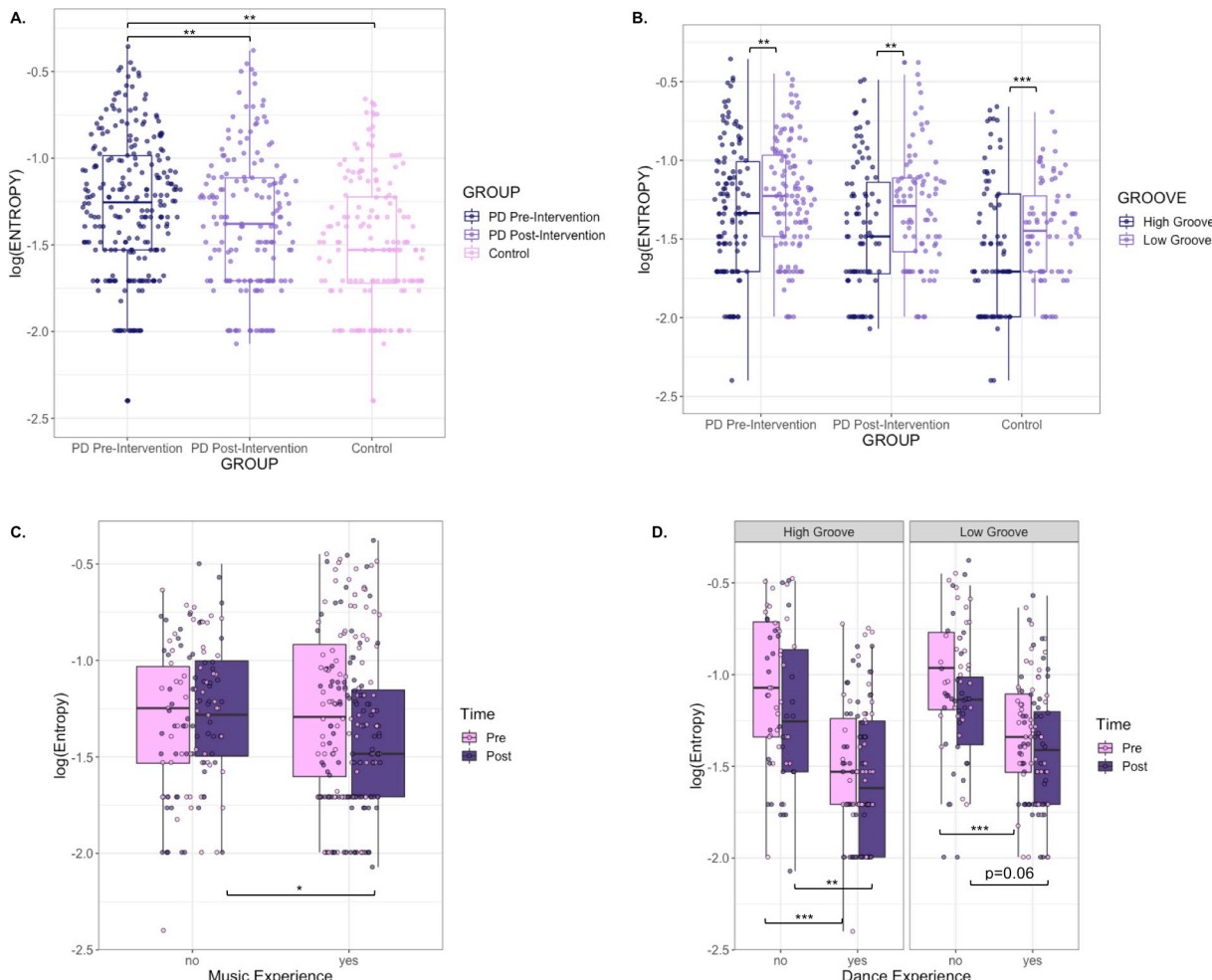

**Fig 5. A.** Comparison of log(Entropy) values for pre-intervention PD participants, post-intervention PD participants, and control participants. **B.** Fig 4A by groove. **C.** Log(Entropy) values by time of testing and music experience. **D.** Pre- and post-intervention log(Entropy) values separated by groove, time of testing, and dance experience.

p = 0.561, had no effect on model fit. F-tests revealed the significant main effect of DE [F(1,27) = 9.33, p = 0.00501**], indicating that those with DE exhibited lower tapping entropy than those without, and secondary effects of Time-Point [F(1,384) = 5.52, p = 0.0193*] and Groove [F(1,366) = 5.36, p = 0.0211*], indicating that entropy for both groove levels decreased post-intervention, but that low-groove entropy was consistently higher than high-groove entropy. F-tests also revealed significant interactions between Time-Point and ME [F(1,384) = 7.44, p = 0.00667**], where post-intervention entropy values were lower than pre-intervention values for participants with ME, but not for those without it [t(334) = -2.30, p = 0.0219*, d = -0.251] (Fig 5C), and between DE and Groove [F(1,366) = 4.77, p = 0.0296*] (Fig 5D).

Post-hoc analysis using Tukey's HSD test revealed significant differences between pre-intervention log(Entropy) values for PD participants with and without DE for both high-groove songs (diff = -0.464, lwr = -0.716, upr = -0.212, p adj = 1.10e-06***, d = 1.21) and low-groove songs (diff = -0.352, lwr = -0.604, upr = -0.100, p adj = 6.84e-04***, d = 1.05). Significant differences were also found between post-intervention log(Entropy) values for PD participants with and without DE for high-groove songs (diff = -0.332, lwr = -0.584, upr = -0.0805, p

adj = 0.00179**, d = 0.847), but not for low-groove songs (diff = -0.246, lwr = -0.498, upr = 0.00558, p adj = 0.0605, d = 0.694) (Fig 5D).

### 3.4 UPDRS improvement is associated with sensorimotor coupling ability

We next assessed the relationship between log(Entropy) and improvements in UPDRS scores (Fig 6A) and found that baseline (pre-intervention) log(Entropy) was positively correlated with z-scores of UPDRS changes for both high-groove (Pearson's correlation: r = 0.302, p = 0.00518**) and low-groove (r = 0.382, p = 0.000336***) songs. Those who tapped with lower entropy during pre-intervention testing thus showed more improvement in PD symptoms. Similar patterns were found in correlations of post-intervention PD log(Entropy) values and z-scores of UPDRS changes (Fig 6B) (high-groove: r = 0.305, p = 0.00498**; low-groove: r = 0.363, p = 0.000683***). Thus, low tapping variability was a significant predictor of reduction in PD symptoms before dance intervention, but not after dance intervention.

We followed up on the relationship between UPDRS change and DE, as well as ME, using paired t-tests. PD participants with DE exhibited significantly greater improvements in UPDRS scores than those without (t(17.9) = 2.80, p = 0.0119*, d = 1.16) (Fig 6C). This difference in UPDRS improvements was not observed when comparing participants with and without ME (t(9.46) = -0.234, p = 0.820, d = 0.121) (Fig 6D). Finally, neither the number of years of ME (r = -0.138, p = 0.670) nor of DE (r = 0.140, p = 0.66) was found to be associated with improvements in UPDRS scores.

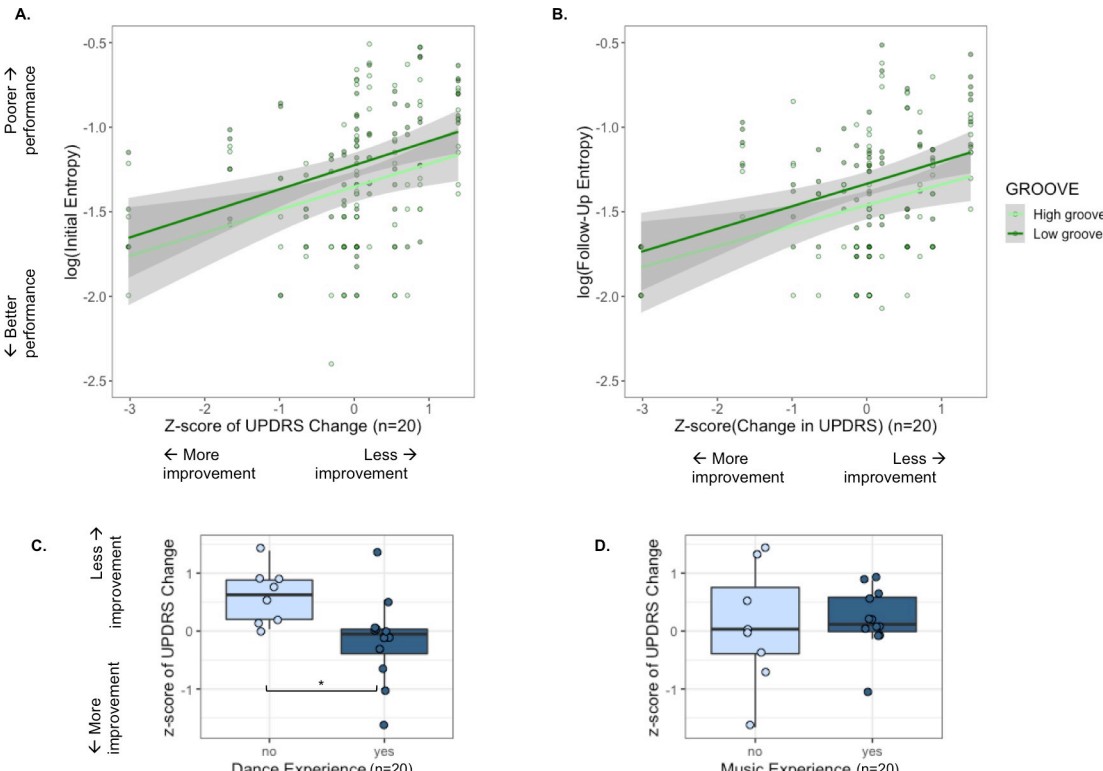

**Fig 6. A.** Relationships between entropy of pre-intervention tapping data and z-scores of changes in UPDRS from baseline to four months, for high-groove and low-groove trials. **B.** Relationships between entropy of post-intervention tapping data and z-scores of changes in UPDRS from baseline to four months, for high-groove and low-groove trials. **C.** Z-scores of changes in UPDRS from baseline to four months for participants with and without dance experience. **D.** Z-scores of changes in UPDRS from baseline to four months for participants with and without music experience.

## 4. Discussion

Results showed that dance classes were associated with reduced PD symptoms, and that dance experience and sensorimotor coupling ability (as assessed by rhythmic tapping) both contributed to the effectiveness of dance classes in reducing PD symptoms. While previous studies have demonstrated that dance classes for PD improve PD symptoms, they have focused primarily on the classes and symptoms themselves through investigating partnered versus non-partnered movement [29], comparisons between different dance styles [51], assessments of gait and balance [23,24,31,52], mood and quality of life measures [24–27], and the effects of dance intervention on specific aspects of movement coordination [32]. To our knowledge, this is the first study to employ objective rhythm tests, to assess previous training in music and dance as predictors of individual differences in responsiveness to PD dance classes, and to incorporate assessments of groove into finger-tapping tests for PD participants and age-matched healthy controls. By comparing participants with and without musical and dance experience, and including measures of beat perception, response to musical groove, and tapping ability, the present study aims not only to assess the outcomes of dance classes on PD symptoms, but also to predict these outcomes using behavioral measures before and after dance intervention.

Our BAT results support previous findings that PD individuals exhibit impaired beat perception and sensitivity when compared with healthy controls [3]. This impairment is likely explained by impaired basal ganglia and motor activity and connectivity, which are involved in beat perception ability [39]. Contrary to previous reports [53], we did not find an effect of prior music experience on sensitivity in beat perception among healthy controls; however, we did find an effect of musical experience on sensitivity in beat perception among PD participants. The different pattern of results between PD and controls may be due to the relatively small sample size of our control group; alternately, it could be due to a relative ceiling effect among controls, as both musically experienced and inexperienced control participants performed well above chance and better than PD participants on the BAT. From these results, it appears that the effects of musical experience only emerged when neural circuitry supporting beat perception is compromised in PD. Prior musical experience may have been neuroprotective in that it equipped the PD participants to better identify changes in beat and rhythm.

Tapping task results also showed experience-dependent patterns, although primarily dependent on dance and not music. As a group, PD participants improved significantly on the tapping task at post-intervention testing; however, this improvement was larger for those without dance experience. This was likely due to better performance among the participants with DE before intervention, as PD participants with DE performed strongly and consistently across time-points. Those without DE performed worse during pre-intervention testing, but improved post-intervention to the point where their high-groove tapping was more similar to those with DE, and there were no longer any significant DE-dependent differences in post-intervention low-groove tapping. Decreased entropy leads to increased predictability of rhythm [54], and because variability in tapping performance is associated with superior sensorimotor coupling ability [55,56], lower tapping entropy in PD participants with prior dance experience may again suggest a neuroprotective effect of dance training. PD participants with music experience similarly demonstrated an improvement in tapping entropy during post-intervention testing compared to pre-intervention testing; this may reflect utilization of the audiomotor circuitry underlying this group's enhanced beat perception ability. And while only the presence of ME and DE—not the duration—seemed to have a significant impact on tapping task performance, further investigation with a larger sample size may reveal relationships between tapping task performance and duration of ME and DE that are not captured by the current study.

DE-dependent performance on tapping was also found to be predictive of post-intervention improvements in UPDRS. Correlations between pre-intervention tapping entropy and changes in UPDRS scores suggest that enhanced—less random, more predictable—sensorimotor coupling ability was associated with larger improvements in disease symptoms. This finding is similar to previous reports associating tapping task performance and therapeutic outcome following RAS in individuals with PD [21]. Increased therapeutic benefit from Parkinson's dance classes in individuals with prior dance training may be reflective of changes in structure and function in brain regions involved in dancing, such as motor and premotor regions and the corticostriatal network [57], which is also disrupted in PD [58]. Dance training—and to perhaps a lesser extent, music training—may strengthen neural pathways involved in motor control, auditory-motor entrainment, action observation, and/or kinesthetic or proprioceptive feedback mechanisms that could then be utilized as an alternative to impaired motor areas. This training-dependent strengthening could in turn prime PD individuals with DE to be more responsive to dance intervention, while being more stable over time in tapping tasks. While participants with DE performed consistently in tapping tasks, participants without DE exhibited more improvement in tapping, which may suggest more plasticity in sensorimotor coupling ability. This pattern of results could reflect multiple mechanisms in which dance helps PD: both as a long-term neuroprotective agent of underlying motor and corticostriatal circuits, and as a shorter-term facilitator or primer of more stable motor behavior.

While DE generally improved tapping as shown in decreased entropy for all trial types among those with DE, ME showed an interaction with time-point in that participants with ME showed more improvement at the post-intervention testing. This may be because ME predisposes individuals to react better to the dance intervention, or it may suggest that those with ME adapt more quickly to the tapping task. The present results could not disentangle between an effect of intervention and an effect of general improvement over the course of repeated testing. Future studies should implement a control intervention to tease apart these competing interpretations.

Despite this complex relationship between pre-intervention tapping stability and therapeutic benefit from dance classes, the indication of learning in PD participants without DE and in participants with ME raises questions about how improvements in sensorimotor coupling may affect responses to dance classes over longer periods of time.

Another factor that may also play a role in explaining our findings is cognitive reserve, i.e. experiential factors that affect individual differences in participants' ability to resist or compensate for behavioral and/or cognitive declines in later life [59], and have been recently associated with cognitive and motor function in PD [60]. In this regard, dance experience may well serve as an experiential factor that builds cognitive reserve, as dance is a cognitively demanding experience that is also a relatively common leisure activity among older adults [61]. Although we did not test this directly, the therapeutic outcome of dance classes may also be associated with socialization and sense of community [62]. Incorporating movement, relaxation, and social interaction with music could further decrease stress and anxiety and, consequently, neuroinflammation and oxidative stress [63–66]. These potential side benefits of dance merit further investigation as they may be used to better tailor dance therapy in the future.

While all PD participants in this study were tested around the same time of day (late morning and midday), in an "on" phase during testing, and on dopaminergic medication for PD, we did not control for individual differences in pharmacological routine. We also did not control for participation in additional therapies beyond the attendance of weekly dance classes because of the potential benefit other therapies may have provided. Because participants were uncomfortable with giving video consent during the study, we were unable to score the UPDRS using blinded videos, which may have introduced an observer bias to the UPDRS.

Nevertheless, results from the tapping data and beat perception tests did not depend on observers or coding. However, there may have been test-retest effects, as participants could have been more comfortable with the tasks the second time around and therefore performed better during post-intervention testing.

Another caveat of these findings is the lack of random assignment and a control intervention: due to limitations in time and resources, we were only able to recruit participants who self-selected into dance intervention, and we were not able to compare dance intervention against a control intervention in this study. Although we had a well-matched control group who were not affected by PD, this control group did not receive intervention, and was smaller in sample size than the PD group. Nevertheless, our healthy control group provided aged-matched control data for the BAT and the rhythmic tapping tasks, which showed better performance (superior beat perception and lower tapping variability) than the pre-intervention but not the post-intervention PD participants. Furthermore, all PD participants had already been attending Dance for PD classes prior to enrolling in our study. Thus, our results may not generalize to a population without ongoing dance training at baseline. Future studies should include assessments prior to the start of the intervention in order to characterize the trajectory of possible improvements due to dance intervention.

Another limitation of the present results is that a third of the PD participants were lost to follow-up, for a variety of reasons as described in Materials and Methods. While concerns about self-selection were partly alleviated by finding similar performance between pre-intervention UPDRS scores for PD participants who did and did not complete the post-intervention tests, the issue of survivor bias remains, in that those who participated in post-intervention testing could have been more engaged, better-supported, or otherwise better situated to improve in symptoms after intervention. While future studies are needed to eliminate the possibility of survivor bias, the current findings are not inconsistent with previous results in showing improvement in motor symptoms following dance intervention [23,24,31,52]. While replicating these results, we also provide additional detailed data in support of rhythm and sensorimotor coupling as the underlying mechanisms that support improvements from dance intervention.

## 5. Conclusions and future directions

The present study tested the effects, and examined mechanistic predictors of success, of a dance intervention for PD. Our findings support the implementation of dance programs in PD communities, as dance classes create a supportive environment that can improve motor ability and decrease symptom severity. Our results also support involvement in dance for healthy individuals, as the presence of previous dance training significantly affected sensorimotor coupling, which in turn predicts the therapeutic outcome of movement therapy for PD.

Through the use of objective rhythm assessments, we found associations between previous dance experience, ability to predictably entrain movements to a musical beat, and improvements in PD symptoms following a dance intervention. Our study has implications for the mechanistic understanding of behavioral interventions for neurodegenerative disorders, and may inform individualized interventions to maximize therapeutic outcome of dance classes for PD. Looking ahead, it would be beneficial to extend the current study to invite more PD communities to participate. Adding a third follow-up session may also further elucidate the groove-related effects on sensorimotor experience, as well as the effects of dance classes on participants without previous dance training. Given the observed benefits of dance for PD in the present study, future studies should test dance interventions more widely to see if the observed benefits persist over time and across larger samples, and if these benefits may also apply to other disease populations.

## Supporting information

**S1 Fig. Changes in UPDRS score in each section (UPDRS I-IV) by dance experience (Dance Exp).** Tukey's HSD post-hoc testing revealed no significant differences between changes in scores by UPDRS section (all p values > 0.05), nor were there significant differences by section for participants with and without DE (all p values > 0.1).
(TIFF)

**S1 Table. BAT stimuli as presented by Iversen and Patel [48].** Each stimulus was designated as either "on the beat" or "off the beat".
(DOCX)

**S2 Table. Musical excerpts used in the tapping task.** Each excerpt has a unique groove rating as assigned by Janata, et al. [41]. The higher four groove ratings were designated as high-groove and the lower four as low-groove.
(DOCX)

## Acknowledgments

Special thanks to Jessica Grahn for helpful advice on the design of this study, to Parker Tichko for help with statistical methods, to Shinya Fujii for helpful comments on data analysis, and to David Leventhal for his support of the project.

## Author Contributions

**Conceptualization:** Anna Krotinger, Psyche Loui.

**Data curation:** Anna Krotinger, Psyche Loui.

**Formal analysis:** Anna Krotinger, Psyche Loui.

**Funding acquisition:** Psyche Loui.

**Investigation:** Anna Krotinger, Psyche Loui.

**Methodology:** Anna Krotinger, Psyche Loui.

**Project administration:** Anna Krotinger.

**Resources:** Anna Krotinger, Psyche Loui.

**Software:** Anna Krotinger, Psyche Loui.

**Supervision:** Psyche Loui.

**Validation:** Psyche Loui.

**Visualization:** Anna Krotinger, Psyche Loui.

**Writing – original draft:** Anna Krotinger.

**Writing – review & editing:** Anna Krotinger, Psyche Loui.

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
