## [Decision Letter · Decision Letter 0]

22 Dec 2020

PONE-D-20-30913

Assessing and Predicting Efficacy of Dance Intervention for Parkinson’s Disease

PLOS ONE

Dear Dr. Loui,

Thank you for submitting your manuscript to PLOS ONE. After careful consideration, we feel that it has merit but does not fully meet PLOS ONE’s publication criteria as it currently stands. Therefore, we invite you to submit a revised version of the manuscript that addresses the points raised during the review process.

In the revised version, please do address the comments of the two reviewers. In the opinion of the Academic Editor, It does not appear that any further analysis or data collection are needed. However, there are significant concerns that should be addressed in the revision. Reviewer 2 has noted that there is insufficient detail regarding the intervention in order to replicate or fully understand the study. This is a major consideration, as work published in PLoS ONE must be replicable and conducted rigorously. Reviewer 1 has noted several interesting points of consideration for the discussion as well as made some very specific suggestions for the tables.

We look forward to receiving your revised manuscript.

Kind regards,

J. Lucas McKay, Ph.D., M.S.C.R.

Academic Editor

PLOS ONE

Journal Requirements:

Reviewers' comments:

Reviewer's Responses to Questions

**Comments to the Author**

1. Is the manuscript technically sound, and do the data support the conclusions?

Reviewer #1: Yes

Reviewer #2: Partly

2. Has the statistical analysis been performed appropriately and rigorously? 

Reviewer #1: Yes

Reviewer #2: Yes

3. Have the authors made all data underlying the findings in their manuscript fully available?

Reviewer #1: Yes

Reviewer #2: Yes

4. Is the manuscript presented in an intelligible fashion and written in standard English?

Reviewer #1: Yes

Reviewer #2: Yes

5. Review Comments to the Author

Reviewer #1: Interesting paper about groove and beat interpretation in response to dance training. The design appears a bit adhoc so the strength of the evidence is a little in question. Many more details about the dance classes should be provided in the Methods section. Currently, all I could find is participants were recruited from various dance classes around the United States. The control group has a lot more dance experience than those with PD. Why are the authors comparing these two groups?

The hit rates in the table could use some metric for units. Same for tapping task, high groove and low groove. I understand if it's 'difficult' to produce a unit, but try to explain what is meant by the score.

the UPDRS III scores indicate these individuals had almost no parkinson's- these are Hoehn and Yahr stage 1 patients. Really? If so, the complications of therapy scores are curious. I would not expect patients with such low scores on the UPDRS III to have much if any medication complications.

Table 1 should be divided into subject characteristics and then outcome variables- make two tables. It is confusing too- the outcome variables initial and follow up- does that mean they are presenting overall means for both groups? Or means for just the PD group? Seems like at least two columns are missing in this table.

Figure 2 title- consider rewording- "separated by..." does not make a lot of sense.

Section 3.3 goes on to include a lot of statistical methods that should be placed in Methods/Statistical analysis section.

Overall - this section is written up well in terms of statistical precision- but please remove anything that is methods and try to be more concise if possible.

The argument that low and high groove represents "sensorimotor coupling ability" needs to be strengthened in the introduction.

Maybe a figure of the concepts- of what tapping is supposed to represent, of what the BAT is supposed to represent or what entropy is supposed to represent would help the reader make their way through the results description.

We know the patients took weekly classes for four months- please give more detail. How many classes did the patients actually take, ie percent adherence, or number of classes taken out of total offered, etc. Summary data would be fine.

The measures (BAT, tapping task) themselves are very interesting and likely of use to those who study dance and music. The study design and lack of clear role of the the control group is a detractor to the paper. If the authors could address these issues, this would strengthen the paper.

Reviewer #2: This study investigated the role of dance practice on motor symptoms of Parkinson’s disease measured using UPDRS, but more importantly the ways in which dance affects rhythmic timing in this population. Results showed that 4-month dance practice showed small effects on motor symptoms and rhythmic timing. The study also considered prior musical experience and showed correlation between musical experience and post-intervention improvement in the UPDRS. The authors conclude that dance classes for PD induce improvement in qualitative and quantitative aspects of PD and that the novel aspect of their study is inclusion of measures of timing.

Main comment

This is a well written paper, investigating an important issue in PD research, namely effects of dance in improving quality of life in people with this disorder. The ideas and questions investigated here are timely and would be useful to researchers in this field. However, there is a major methodological issue with the study.

The issue is the dance intervention itself. There is no clear description of the nature of the dance intervention, i.e. the content of the dance classes. The study is very clear in reporting the tests performed pre- and post-intervention, but mentions nothing in the methods section about the intervention itself. What kind of dance classes were these? There are many different ways to run a dance class for people with PD. There is folk dance, tango (see work by Hackney and colleagues) and Dance for PD which is widely used in many countries. The authors mention three sites in different parts of the US in which they collected data. Did all three sites run the same intervention, or did each site run different classes? Were these classes coordinated/controlled by the experimenters, in terms of their content and methods? Finally, how many classes did participants attend on average? If we don’t know what these classes were, and if we don’t know if they all did the same thing it is very hard to interpret these results.

Another issue is that half of the participants who finished the intervention had been participating in dance classes for some time before the pre-test. That means that this is not a real pre-test, in the sense that you are not measuring what happened before the intervention, the intervention was going on before the pre-test as well. However, what is puzzling is that one would expect greater effects of the intervention at the UPDRS for people with no dance experience but instead the opposite was found to be the case. The authors could discuss this in the discussion section.

There are other, more minor methodological issues with the study, but I mention them as minor because the authors acknowledge them in the discussion as limitations, for example issues with the control group and the drop in the number of participants post-test. These, and other of the limitations art part and parcel of this type of research which is extremely hard to perform. However, the content of the classes and its consistency cross cites is a major issue.

There are two recent reviews on the topic that the authors could include, one by Bek et al (2020), NNR; and one by Carapellotti (2020) in PLoS one.

6. PLOS authors have the option to publish the peer review history of their article (what does this mean?). If published, this will include your full peer review and any attached files.

Reviewer #1: No

Reviewer #2: No

---

## [Author Response · Author response to Decision Letter 0]

28 Jan 2021

Dear Dr. McKay,

Thank you for handling our manuscript, “Assessing and Predicting Efficacy of Dance Intervention for Parkinson’s Disease,” for possible publication in PLoS ONE. We also thank the reviewers for their careful feedback and helpful comments. We now submit a revised version, with substantial additions about the details of the dance intervention, the inclusion of the control group in our design, and a new Figure 1 that provides a conceptual overview of the assessments used. Point-by-point responses to the reviewers’ comments are below. We hope that you might find this version to be suitable for publication for the wide readership of PLoS ONE.

Sincerely,

Psyche Loui

Anna Krotinger

Reviewer #1: 

1. Many more details about the dance classes should be provided in the Methods section. Currently, all I could find is participants were recruited from various dance classes around the United States. 

We thank the reviewer for pointing this out and we welcome the opportunity to expand on the details of the dance classes. All dance classes were taught by instructors who had received teaching certification from Dance for PD®. While the specific content of each class changed weekly, the structure was consistent and the primary form of movement was in a contemporary/modern style. Each class lasted 75 minutes and was set to live music. About 40 minutes were spent seated. While seated, instructors led exercises emphasizing muscle and limb extension and rhythmic movement. These exercises were typically broken into combinations focused on coordination (dancers might be instructed to clap on certain counts, or to alternate clapping and stomping), footwork (involving leg extension, flexing and pointing the feet, and raising heels to a forced arch position, for example), and torso (focusing on leaning forward and side to side, stretching the sides of the body, and coordinating arm movements with other motions), and explore a variety of musical tempos and rhythms. For the remainder of the class, instructors invited participants to stand if they were able. For about twenty minutes, participants performed exercises standing at the barre or holding on to the back of their chairs. These combinations emphasized plié, or bending and straightening the knees; lifting the heels off the ground to balance; and extending the legs and feet in different directions. Finally, participants moved across the floor for the last fifteen minutes of class, performing exercises involving weight shifting, marching, partnering (participants mirror others’ movement), and improvisation. In preparation for each combination, the instructor demonstrated a sequence of movements and then provided spoken cues while all participants performed the combination to music. Each class had at least one volunteer who provided dancers with support and assistance.

The above information is now included in Methods in a section titled “2.1.2 Intervention.”

2. The control group has a lot more dance experience than those with PD. Why are the authors comparing these two groups?

Thank you for the opportunity to elaborate on our design, which included one session of testing non-PD control participants, as well as two sessions (pre- and post-intervention) of testing PD participants. While the tapping task with different levels of groove is a sensitive measure of rhythm and sensorimotor coupling, at present we know of no normed data from these assessments within the age range of those in our PD sample. Thus we believe it was necessary to validate these behavioral assessments against an age-matched control group without PD, within the scope of the present study.

Control participants were recruited from caretakers/close associates/family members, and thus were a sample of convenience that was age-matched to the PD group. 66.7% of PD participants and 63.1% of control participants had received at least one year of formal dance training. The reviewer is correct that the average years of dance training for control participants was about two times that of PD participants. However, neither PD nor control subjects’ durations of DE are normally distributed [see figure below for histogram of Years of DE by Group (PD and Control)]. Here, we simply categorized participants by the presence/absence of DE to assess the impact of DE on participants’ performance on objective rhythm tests. 

Histogram of Years of DE for PD and Control groups.

In investigating the impact of discrepancies in years of DE on tapping task results, though, we did find associations between the number of years of dance experience and Shannon entropy for some participants. We found a negative correlation between post-intervention PD participants’ tapping task entropy and years of dance experience (Pearson’s correlation: r = -0.248, p = 0.011*). However, correlations between both pre-intervention PD entropy values and years of DE (r = -0.150, p = 0.0648) and control participants’ entropy values and years of DE (r = -0.190, p = 0.0640) did not reach significance. 

To tease apart the effect of Years of DE from the effect of group (PD vs. control) on tapping entropy, we included “Years of DE” as a fixed effect in all LME models comparing control and PD Shannon entropy. For LMEs comparing pre-intervention PD and control log(Entropy) values, likelihood ratio tests revealed that adding Group, χ2(1) = 7.51, p = 0.00612** and Groove, χ2(1) = 9.48, p = 0.00208** significantly and independently improved model fit. Adding Years of DE, χ2(11) = 17.1, p = 0.104, did not improve model fit. F-tests confirmed the main effects of Group [F(1,47) = 7.79, p = 0.00757**] and Groove [F(1,342) = 9.58, p = 0.00213**], indicating that pre-intervention PD log(Entropy) values were significantly higher than control values, and again that high-groove entropy values were lower than low-groove values. All interaction terms were non-significant (all p values > 0.2).

LMEs comparing post-intervention PD and control log(CV) values revealed that including Groove, χ2(1) = 12.6, p = 0.000393***, but not Group, χ2(1) = 1.88, p = 0.17, significantly improved model fit. F-tests revealed a significant main effect of Groove [F(1,279) = 12.8, p = 0.000408***], indicating again that high-groove entropy values were lower than low groove values. While including Years of DE improved model fit, χ2(11) = 22.2, p = 0.0228*, an F-test failed to reject the null hypothesis that Years of DE had no independent effect on log(Entropy) [F(1,11) = 1.89, p = 0.0853]. All interaction terms were non-significant (all p values > 0.2). Together, these results demonstrate that PD participants reduced their tapping variability after the intervention, with tapping entropy more similar to that of controls. Furthermore, while tapping entropy varied across songs, high-groove entropy was consistently lower than low-groove entropy, and duration of DE did not interact with these effects.

These updated LME models above, with Years of DE as a fixed effect, have been incorporated into Section 3.3 of our manuscript. We have also noted in the discussion that with a larger sample size, further investigation may reveal relationships between tapping task performance and duration of DE that are not captured by the current study. While we thank the reviewer for pointing out that the controls had more years of DE on average than our PD group, we hope these new results assuage any potential concerns of the between-group effects being confounded by Years of DE.

3. The hit rates in the table could use some metric for units. Same for tapping task, high groove and low groove. I understand if it's 'difficult' to produce a unit, but try to explain what is meant by the score.

We thank the reviewer for pointing this out and agree that more explication would help clarify the data. In our original manuscript, we expressed hit rate as the proportion of correctly detected differences given the overall number of differences, whereas false alarm rate was the proportion of trials where the response was “different” given that the correct response was “same.” In this revision, we express these values as percentages. 

For the tapping task, we have also clarified that the values listed for the tapping task reflect Shannon entropy of phase angles. While we understand the reviewer’s ask for units for entropy, some sources cite “Shannon” as the unit but in this case that unit is less appropriate because it applies to probabilities of phase angles, rather than a more classical linear measure. Since entropy is by nature a proportional value, in this revision we explain these values by giving a more in-depth conceptualization of these measures. We also provide the theoretical upper and lower bounds of entropy for reference. In section 2.3.3, we now clarify that Shannon entropy reaches zero if the probability of a certain IOI value (or phase angle) is 1. This gives us the possible range of entropy values: because we used 400 bins to construct polar histograms of each tapping trial, the highest possible entropy is -(400*1/400*log(1/400) = 2.602, indicating complete randomness, while the lowest is 0, indicating robotic tapping.

4. The UPDRS III scores indicate these individuals had almost no parkinson's- these are Hoehn and Yahr stage 1 patients. Really? If so, the complications of therapy scores are curious. I would not expect patients with such low scores on the UPDRS III to have much if any medication complications.

The reviewer is correct that these patients mostly fall in H&Y Stage 1. However, 

we note that the UPDRS IV scores are very low across the board. In a study assessing the validity of UPDRS Section IV, Raciti et al. (2016), found that the average UPDRS IV score for PD patients who did not experience motor fluctuations--and thus experienced few, if any, medication complications--was 2.87 ± 2.4. For patients experiencing motor fluctuations, the UPDRS IV scores averaged 5.7 ± 2.6. . Our data thus converge with the reviewer’s expectation that low UPDRS III scores would beget few, if any, complications of therapy. Out of a possible 23 points in UPDRS IV, the average score for pre-intervention participants in our study was 3.8 ± 3.84, and post-intervention, 3.05 ± 3.09. These data reflect a participant pool in which most, but not all PD participants experience very few medication complications, consistent with the reviewer’s expectation. 

Raciti, L., Nicoletti, A., Mostile, G., Bonomo, R., Contrafatto, D., Dibilio, V., . . . Zappia, M. (2016). Validation of the UPDRS section IV for detection of motor fluctuations in Parkinson's disease. Parkinsonism Relat Disord, 27, 98-101. doi:10.1016/j.parkreldis.2016.03.008

5. Table 1 should be divided into subject characteristics and then outcome variables- make two tables. It is confusing too- the outcome variables initial and follow up- does that mean they are presenting overall means for both groups? Or means for just the PD group? Seems like at least two columns are missing in this table.

Table 1 has been separated into two tables as indicated. We have also clarified that there are three columns for the outcome variables table: PD pre-intervention, PD post-intervention, and control values. Control participants do not have pre- and post-intervention values, so are listed in a single column. 

6. Figure 2 title- consider rewording- "separated by..." does not make a lot of sense.

We agree that the initial wording was confusing. We have now updated the text to read, “With and without music experience (ME) and dance experience (DE)” instead of “separated by…”

7. Section 3.3 goes on to include a lot of statistical methods that should be placed in Methods/Statistical analysis section. Overall - this section is written up well in terms of statistical precision- but please remove anything that is methods and try to be more concise if possible.

Thank you for pointing this out. We have removed all description of LME model construction from Section 3.3 and have included it in Section 2.3.3 (Methods: Statistical analysis). Section 3.3 is now more concise. 

8. The argument that low and high groove represents "sensorimotor coupling ability" needs to be strengthened in the introduction.

We agree with the reviewer that the introduction could use more explanation on the role of groove in sensorimotor coupling. Evidence for the role of groove in sensorimotor coupling comes from results involving transcranial magnetic stimulation (TMS) of the motor cortex: when asked to tap along to different songs, healthy participants show the most spontaneous motor excitability, as indicated by motor evoked potentials in response to TMS, in response to high-groove songs as compared with songs categorized as low-groove. This suggests that groove mediates the way auditory rhythms excite the motor system; in other words, groove drives sensorimotor coupling. This is now added to the Introduction.

9. Maybe a figure of the concepts- of what tapping is supposed to represent, of what the BAT is supposed to represent or what entropy is supposed to represent would help the reader make their way through the results description.

We agree with the reviewer that a visual / conceptual overview would be helpful. In this revision we add a new Figure 1 to show our conceptualizations of the problem of dance experience and intervention, the cognitive constructs and neural systems that they tap into, and the tools that we use to assess these constructs. We thank the reviewer for this suggestion and we hope that it clarifies our thinking on dance for PD. 

Fig 1. Conceptual overview of relationships between relevant cognitive and behavioral assessments, constructs, and activities in the context of Parkinson’s disease.

10. We know the patients took weekly classes for four months- please give more detail. How many classes did the patients actually take, ie percent adherence, or number of classes taken out of total offered, etc. Summary data would be fine.

In addition to what was said in response to Point 1, we now include additional information on how compliance with the intervention affected inclusion/exclusion criteria. The minimum requirement for dance class attendance, then, was one class per week, or 16 total classes. If by the post-intervention meeting (four months after the initial meeting) participants had failed to attend at least one class per week, their data were excluded from follow-up analysis. We did not exclude participants for additional class attendance, or for participating in any other interventions. We have now included clarification of the 16 class requirement (one per week) in Section 2.1.4.

11. The measures (BAT, tapping task) themselves are very interesting and likely of use to those who study dance and music. The study design and lack of clear role of the control group is a detractor to the paper. If the authors could address these issues, this would strengthen the paper.

Thank you for your comment. We have now added more of a conceptual overview, including by adding a new Figure 1. We refer to the BAT and tapping task, along with other behavioral and cognitive components of our study, in the introduction and in Figure 1. We now also add a rationale for use of the control group. To validate the relatively novel use of these tasks to assess rhythm capabilities and sensorimotor coupling, we needed an additional control group who had no PD to serve as a comparisonator for PD participants’ performance on objective rhythm tasks at both timepoints 

Reviewer #2: 

1. There is no clear description of the nature of the dance intervention, i.e. the content of the dance classes. The study is very clear in reporting the tests performed pre- and post-intervention, but mentions nothing in the methods section about the intervention itself. What kind of dance classes were these? There are many different ways to run a dance class for people with PD. There is folk dance, tango (see work by Hackney and colleagues) and Dance for PD which is widely used in many countries. The authors mention three sites in different parts of the US in which they collected data. Did all three sites run the same intervention, or did each site run different classes? Were these classes coordinated/controlled by the experimenters, in terms of their content and methods? Finally, how many classes did participants attend on average? If we don’t know what these classes were, and if we don’t know if they all did the same thing it is very hard to interpret these results.

All dance classes were taught by instructors who had received teaching certification from Dance for PD®. While the specific content of each class changed weekly, the structure was consistent and the primary form of movement was in a contemporary/modern style. Each class lasted 75 minutes and was set to live music. About 40 minutes were spent seated. While seated, instructors led exercises emphasizing muscle and limb extension and rhythmic movement. These exercises were typically broken into combinations focused on coordination (dancers might be instructed to clap on certain counts, or to alternate clapping and stomping), footwork (involving leg extension, flexing and pointing the feet, and raising heels to a forced arch position, for example), and torso (focusing on leaning forward and side to side, stretching the sides of the body, and coordinating arm movements with other motions), and explore a variety of musical tempos and rhythms. For the remainder of the class, instructors invited participants to stand if they were able. For about twenty minutes, participants performed exercises standing at the barre or holding on to the back of their chairs. These combinations emphasized plié, or bending and straightening the knees; lifting the heels off the ground to balance; and extending the legs and feet in different directions. Finally, participants moved across the floor for the last fifteen minutes of class, performing exercises involving weight shifting, marching, partnering (participants mirror others’ movement), and improvisation. In preparation for each combination, the instructor demonstrated a sequence of movements and then provided spoken cues while all participants performed the combination to music. Each class had at least one volunteer who provided dancers with support and assistance.

In addition to what was said in response to Point 1, here is some additional information on how the compliance with the intervention affected inclusion/exclusion criteria. If by the post-intervention meeting (four months after the initial meeting) participants had failed to attend at least one class per week, their data were excluded from follow-up analysis. The minimum requirement for dance class attendance, then, was one class per week, or 16 total classes. We did not control for additional class attendance or other interventions, nor did we collect that information.

This information is now included in Methods in a section titled “2.1.2 Intervention,” as well as in a response to Reviewer 1’s Points 1 and 10.

2. Another issue is that half of the participants who finished the intervention had been participating in dance classes for some time before the pre-test. That means that this is not a real pre-test, in the sense that you are not measuring what happened before the intervention, the intervention was going on before the pre-test as well. However, what is puzzling is that one would expect greater effects of the intervention at the UPDRS for people with no dance experience but instead the opposite was found to be the case. The authors could discuss this in the discussion section.

We acknowledge that the pre-test did not represent a true baseline in the sense that all PD participants in the study had been previously attending PD dance classes. However, we believe that the four month period does capture some contrast against an earlier stage in participants’ experience with PD dance classes, as demonstrated by the fact that PD participants in our study exhibited symptom improvement, as well as the finding that participants with dance experience exhibited greater improvements. These results point to a need for further investigation into the neuroprotective effects of dance early in life. As you mention, this would ideally be done at any earlier point in disease onset in order to establish a true baseline. Our research suggests that dance experience makes you more susceptible to benefits of dance intervention later on, which is a point that we hope is addressed with future research.

3. There are other, more minor methodological issues with the study, but I mention them as minor because the authors acknowledge them in the discussion as limitations, for example issues with the control group and the drop in the number of participants post-test. These, and other of the limitations are part and parcel of this type of research which is extremely hard to perform. However, the content of the classes and its consistency across sites is a major issue.

We appreciate your acknowledgment of the challenges inherent to this kind of research and hope that we have sufficiently addressed your comments about the content of the classes and their consistency.

4. There are two recent reviews on the topic that the authors could include, one by Bek et al (2020), NNR; and one by Carapellotti (2020) in PLoS one.

Thank you for bringing our attention to these reviews. We have now included reference to both reviews in the introduction, and to Carapellotti et al., in the discussion.

---

## [Decision Letter · Decision Letter 1]

17 Mar 2021

PONE-D-20-30913R1

Assessing and Predicting Efficacy of Dance Intervention for Parkinson’s Disease

PLOS ONE

Dear Dr.Loui,

Thank you for submitting your manuscript to PLOS ONE. We feel that you have addressed the predominant concerns of the reviewers, and that only a relatively small number of changes are required to ensure reproducibility and to adequately convey the study design. These are summarized below.

Academic Editor

Due to the sophistication of the analytic methods used, please provide the specific R function used to fit linear mixed models (stats::glm, LmerTest::lme4, etc.), the function used to perform likelihood ratio tests (anova, drop1(lm1, test="F")), and whether maximum likelihood (ML) or restricted maximum likelihood (REML) was used for model fitting. Without these details readers are unable to critically evaluate results, which can vary somewhat depending on the values chosen.

Reviewer 1

Please alter the title to be more specific than "Dance," as this is over-broad. The reviewer suggests a phrase such as "Contemporary Dance."

Optionally, the reviewer also suggests changing the title to include something more about music, groove and/or rhythm, which are "by far the most interesting aspect of this paper."

Please specify the study design in some way, as the current title gives the impression that this is an RCT. This could be "Efficacy of a Contemporary Dance Intervention for Parkinson’s Disease: an Observational Study"

Reviewer 2

Please specify in the limitations that the results may not generalize to a study population without ongoing dance training at baseline, and that you would either expect increased or decreased efficacy in such a population.

We look forward to receiving your revised manuscript.

Kind regards,

J. Lucas McKay, Ph.D., M.S.C.R.

Academic Editor

PLOS ONE

Journal Requirements:

Reviewers' comments:

Reviewer's Responses to Questions

**Comments to the Author**

1. If the authors have adequately addressed your comments raised in a previous round of review and you feel that this manuscript is now acceptable for publication, you may indicate that here to bypass the “Comments to the Author” section, enter your conflict of interest statement in the “Confidential to Editor” section, and submit your "Accept" recommendation.

Reviewer #1: All comments have been addressed

Reviewer #2: All comments have been addressed

2. Is the manuscript technically sound, and do the data support the conclusions?

Reviewer #1: Yes

Reviewer #2: Yes

3. Has the statistical analysis been performed appropriately and rigorously? 

Reviewer #1: Yes

Reviewer #2: Yes

4. Have the authors made all data underlying the findings in their manuscript fully available?

Reviewer #1: Yes

Reviewer #2: Yes

5. Is the manuscript presented in an intelligible fashion and written in standard English?

Reviewer #1: Yes

Reviewer #2: Yes

6. Review Comments to the Author

Reviewer #1: Thanks for addressing my critique. The paper is strengthened.

I have a suggestion that I think is pretty important- change the title to something more about music, groove and/or rhythm. This aspect is by far the most interesting aspect of this paper. You can also include "Dance for PD, a contemporary dance style" or "contemporary dance in the title rather than simply "dance". I am not sure about 'efficacy' either.

also- we need to see what the editor thinks but the design of the project is important too, given this is not a RCT or a pilot study- it's more a convenience sample, or maybe something "quasi -experimental". I think including that in the title or somewhere in the abstract and methods is important.

Reviewer #2: I have reviewed the revision to the manuscript. the authors have addressed most of my concerns. My final, very minor comment is to ask authors to include their response to my second point which relates to the pre-test as a limitation to the study. The authors acknowledge that "the pre-test did not represent a true baseline in the sense that all PD participants in the study had been previously attending PD dance classes." I would suggest to the authors to include this point as one of the study's limitations.

7. PLOS authors have the option to publish the peer review history of their article (what does this mean?). If published, this will include your full peer review and any attached files.

Reviewer #1: No

Reviewer #2: No

---

## [Author Response · Author response to Decision Letter 1]

24 Mar 2021

Academic Editor:

1. Due to the sophistication of the analytic methods used, please provide the specific R function used to fit linear mixed models (stats::glm, LmerTest::lme4, etc.), the function used to perform likelihood ratio tests (anova, drop1(lm1, test="F")), and whether maximum likelihood (ML) or restricted maximum likelihood (REML) was used for model fitting. Without these details readers are unable to critically evaluate results, which can vary somewhat depending on the values chosen.

Thank you for pointing this out! We have updated section 2.3.3 to read:

“Linear mixed-effects models (LMEs) were constructed using the R function LmerTest::lme4 to investigate whether Groove (High vs. Low), Group (PD vs. Control), and their interaction affected IOI values. First, a base model was constructed using IOI as the dependent variable and participant ID as a random effect. Then, Groove and Group were added as fixed effects. Maximum likelihood (ML) was used for model fitting. Models were compared via likelihood ratio tests using the anova function. All code for analyses and figures are freely available on https://doi.org/10.6084/m9.figshare.13034165.v1.”

Reviewer #1: 

1. I have a suggestion that I think is pretty important- change the title to something more about music, groove and/or rhythm. This aspect is by far the most interesting aspect of this paper. You can also include "Dance for PD, a contemporary dance style" or "contemporary dance in the title rather than simply "dance". I am not sure about 'efficacy' either.

also- we need to see what the editor thinks but the design of the project is important too, given this is not a RCT or a pilot study- it's more a convenience sample, or maybe something "quasi -experimental". I think including that in the title or somewhere in the abstract and methods is important.

We thank the reviewer for pointing out that the title could be improved by including reference to groove and rhythm. We have altered it to read:

“Rhythm and Groove as Cognitive Mechanisms of Dance Intervention in Parkinson’s Disease”

We hesitate to specify the dance intervention as exclusively “contemporary” in the title of the study. In our description of the intervention in section 2.1.2, we describe the intervention thus: “While the specific content of each class changed weekly, the structure was consistent and the primary form of movement was in a contemporary/modern style.” Had the movement in every class been exclusively contemporary, we would feel comfortable noting this in the title. However, each teacher has license to draw influence from other styles of dance, so even if the majority of combinations in a given class are in a contemporary style, there might be a single piece of choreography that draws on African or Latin dance styles, for example. We have clarified this in the body of the manuscript (2.1.2):

“While the specific content of each class changed weekly and teachers were free to draw on different dance styles (e.g. African or Latin) while planning choreography, the structure was consistent and the primary form of movement was in a contemporary/modern style.”

Reviewer #2: 

1. Please specify in the limitations that the results may not generalize to a study population without ongoing dance training at baseline, and that you would either expect increased or decreased efficacy in such a population.

We agree with the reviewer that results may differ if dance training had not been ongoing at baseline. We have now added to the limitations section of Discussion to read:

“Another caveat of these findings is the lack of random assignment and a control intervention: due to limitations in time and resources, we were only able to recruit participants who self-selected into dance intervention, and we were not able to compare dance intervention against a control intervention in this study. Although we had a well-matched control group who were not affected by PD, this control group did not receive intervention, and was smaller in sample size than the PD group. Nevertheless, our healthy control group provided aged-matched control data for the BAT and the rhythmic tapping tasks, which showed better performance (superior beat perception and lower tapping variability) than the pre-intervention but not the post-intervention PD participants.

Furthermore, all PD participants had already been attending Dance for PD classes prior to enrolling in our study. Thus, our results may not generalize to a population without ongoing dance training at baseline. Future studies should include assessments prior to the start of the intervention in order to characterize the trajectory of possible improvements due to dance intervention.”

---

## [Editor Report · Decision Letter 2]

29 Mar 2021

Rhythm and Groove as Cognitive Mechanisms of Dance Intervention in Parkinson’s Disease

PONE-D-20-30913R2

Dear Dr. Loui,

We are pleased to inform you that all reviewer comments have been adequately addressed and that we are happy for the opportunity to publish this unique work. The academic editor extends his personal thanks for your comprehensive attention to comments.

Kind regards,

J. Lucas McKay, Ph.D., M.S.C.R.

Academic Editor

PLOS ONE
---

## [Editor Report · Acceptance letter]

1 Apr 2021

PONE-D-20-30913R2 

Rhythm and Groove as Cognitive Mechanisms of Dance Intervention in Parkinson’s Disease 

Dear Dr. Loui:

I'm pleased to inform you that your manuscript has been deemed suitable for publication in PLOS ONE. Congratulations! Your manuscript is now with our production department. 

Kind regards, 

on behalf of

Dr. J. Lucas McKay 

Academic Editor

PLOS ONE